# Intranasal infection and contact transmission of Zika virus in guinea pigs

Yong-Qiang Deng[1,2], Na-Na Zhang[1], Xiao-Feng Li[1,2], Ya-Qing Wang[3,4], Min Tian[5], Ye-Feng Qiu[6], Jun-Wan Fan[3], Jia-Nan Hao[1,7], Xing-Yao Huang[1], Hao-Long Dong[1], Hang Fan[2], Yu-Guang Wang[5], Fu-Chun Zhang[8], Yi-Gang Tong [2], Zhiheng Xu [3,4] & Cheng-Feng Qin [1,2,8]

Zika virus (ZIKV) is primarily transmitted to humans through mosquito bites or sexual contact. The excretion and persistence of contagious ZIKV in various body fluids have been well documented in ZIKV patients; however, the risk of direct contact exposure remains unclear. Here, we show that guinea pigs are susceptible to ZIKV infection via subcutaneous inoculation route; infected guinea pigs exhibit seroconversion and significant viral secretion in sera, saliva, and tears. Notably, ZIKV is efficiently transmitted from infected guinea pigs to naïve co-caged animals. In particular, intranasal inoculation of ZIKV is fully capable of establishing infection in guinea pigs, and viral antigens are detected in multiple tissues including brain and parotid glands. Cynomolgus macaques also efficiently acquire ZIKV infection via intranasal and intragastric inoculation routes. These collective results from animal models highlight the risk of exposure to ZIKV contaminants and raise the possibility of close contact transmission of ZIKV in humans.

[1] Department of Virology, Beijing Institute of Microbiology and Epidemiology, Beijing 100071, China. [2] State Key Laboratory of Pathogen and Biosecurity, Beijing 100071, China. [3] State Key Laboratory of Molecular Developmental Biology, CAS Center for Excellence in Brain Science and Intelligence Technology, Institute of Genetics and Developmental Biology, Chinese Academy of Sciences, Beijing 100101, China. [4] Parkinson's Disease Center, Beijing Institute for Brain Disorders, Beijing 100101, China. [5] Beijing Traditional Chinese Medicine Hospital, Capital Medical University, Beijing 100010, China. [6] Laboratory Animal Center, Academy of Military Medical Science, Beijing 100071, China. [7] Anhui Medical University, Hefei 230032, China. [8] Guangzhou Eighth People's Hospital, Guangzhou Medical University, Guangzhou 510060, China. Yong-Qiang Deng, Na-Na Zhang, Xiao-Feng Li, Ya-Qing Wang, Min Tian and Ye-Feng Qiu contributed equally to this work. Correspondence and requests for materials should be addressed to Z.X. (email: zhxu@genetics.ac.cn) or to C.-F.Q. (email: qincf@bmi.ac.cn)

The re-emergence of Zika virus (ZIKV) has raised global concerns since 2016 due to its rapid spread worldwide and clinical manifestations in humans. ZIKV is a member of the *Flavivirus* genus in the *Flaviviridae* family, which includes a large number of vector-borne pathogens, such as dengue virus (DENV), West Nile virus (WNV), Japanese encephalitis virus (JEV), yellow fever virus (YFV), and tick-borne encephalitis virus (TBEV), etc. Similar to other mosquito-borne flaviviruses, ZIKV is primarily transmitted to humans through bites by mosquito vectors. However, the vertical transmission of ZIKV from infected pregnant women to their fetuses has been well documented during recent epidemics, resulting in thousands of cases of microcephaly and congenital malformations in newborn infants[1,2].

Recently, clinical and laboratory findings have established sexual contact as an unusual route for human-to-human transmission of ZIKV[3–5]. However, sexual intercourse involves close physical contact and a complex exchange of body fluids, including vaginal secretions, semen, and saliva; therefore, sexual transmission may be difficult to distinguish from other potential transmission routes. Importantly, infectious ZIKV has been directly recovered from various body fluids, including semen, saliva, urine, and breast milk[6–8], and viral RNAs have been detected in nasopharyngeal swabs[9]. The sustained secretion of ZIKV in saliva, urine, and lacrimal fluids has also been observed in symptomatic or nonsymptomatic human or non-human primates[10,11]. Therefore, close contact or exposure to these infectious body fluids through the oronasal mucosa could cause infection; this capability has been well established for other sexually transmitted viruses. A recent report has demonstrated that JEV, a flavivirus homologous to ZIKV, can be transmitted by direct contact via oronasal secretions in pigs[12]. In addition, some other flavivirus members, including WNV and duck Tembusu virus, can be transmitted among geese and ducks via direct contact[13,14]. This phenomenon encouraged us to investigate the transmissibility of ZIKV via direct contact among animals. Notably, a recent case report[15] describing ZIKV infection with no known mosquito bites or sexual contact has increased the urgency of investigating whether ZIKV has been transmitted via other unknown routes during recent epidemics in humans.

In present study, we established our own guinea pig model of ZIKV infection via subcutaneous inoculation route, which exhibited significant viremia and robust viral secretion in saliva and tears. Importantly, we found that ZIKV can efficiently transmit from infected guinea pigs to their naïve co-caged mates. In particular, ZIKV introduced via intranasal (i.n.) inoculation was fully capable of establishing acute infection in guinea pigs, which was closely associated with robust viral replication in the parotid glands. Remarkably, cynomolgus macaques efficiently acquired ZIKV via i.n. and intragastric (i.g.) inoculation routes. The results from animal models highlight the risk of exposure to ZIKV contaminants and raise the possibility of close contact transmission of ZIKV in humans.

## Results

**Guinea pigs are susceptible to ZIKV infection.** To establish a model for mimicking the potential close contact transmission of ZIKV, we first assessed the infectivity of ZIKV in guinea pigs, which have been well characterized and widely used for modeling contact transmission of pathogenic viruses[16–19]. Groups of four adult male guinea pigs were inoculated subcutaneously (s.c.) with varying doses of a 2016 contemporary ZIKV strain from Venezuela[6], and viremia in each animal was monitored thereafter. A dose-dependent infection was observed in all inoculated guinea pigs: 100%, 75%, and 25% of the animals inoculated with $10^5$, $10^4$, and $10^3$ plaque-forming unit (PFU) of ZIKV developed viremia, while no viremia was detected in animals inoculated with $10^2$ PFU of ZIKV; the median infectious dose ($MID_{50}$) of ZIKV in guinea pigs was calculated to $10^{3.5}$ PFU accordingly (Supplementary Fig. 1).

Then, the clinical, virological, pathological, and immunological response in guinea pigs upon $10^5$ PFU challenge were characterized in details. All inoculated animals maintained their weight,

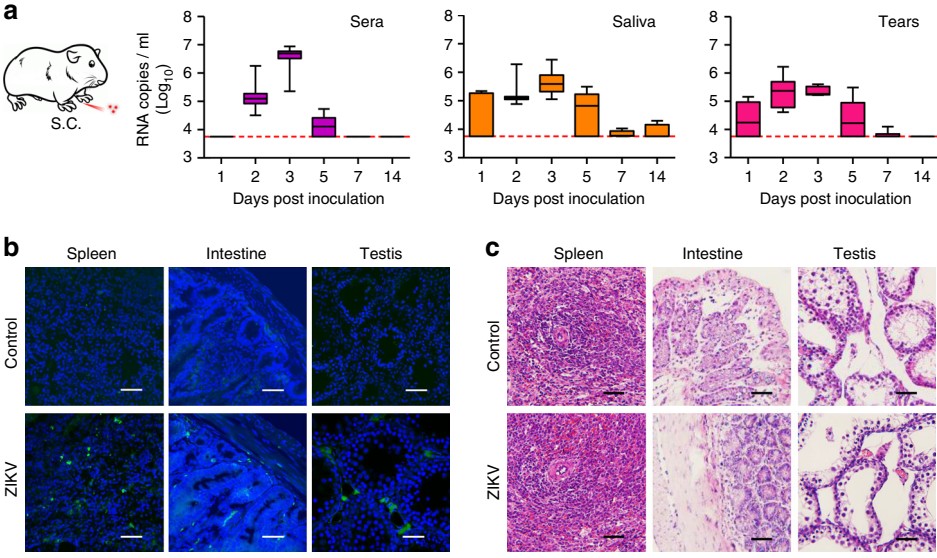

**Fig. 1** Guinea pigs are susceptible to ZIKV infection. **a** Group of adult guinea pigs were s.c. inoculated with ZIKV and viral loads in the serum, saliva, and tear samples were determined using RT-qPCR at the indicated times. The number of infected animals evaluated on each day as follows: days 1 to 3, $n = 8$; days 5 to 7, $n = 6$; day 14, $n = 4$. Viral loads are expressed as RNA copies per milliliter. Dotted lines indicate the limit of detection. Whiskers: 5–95 percentile. **b** Immunostaining were performed with the convalescent serum from a ZIKV patient as the primary antibody. All tissue samples were collected from ZIKV-infected guinea pigs at 3 dpi. Nuclei were stained with DAPI. Scale bar: 50 μm. **c** Histopathological changes in selected tissues from the ZIKV-infected guinea pigs at 3 dpi. Scale bar: 50 μm

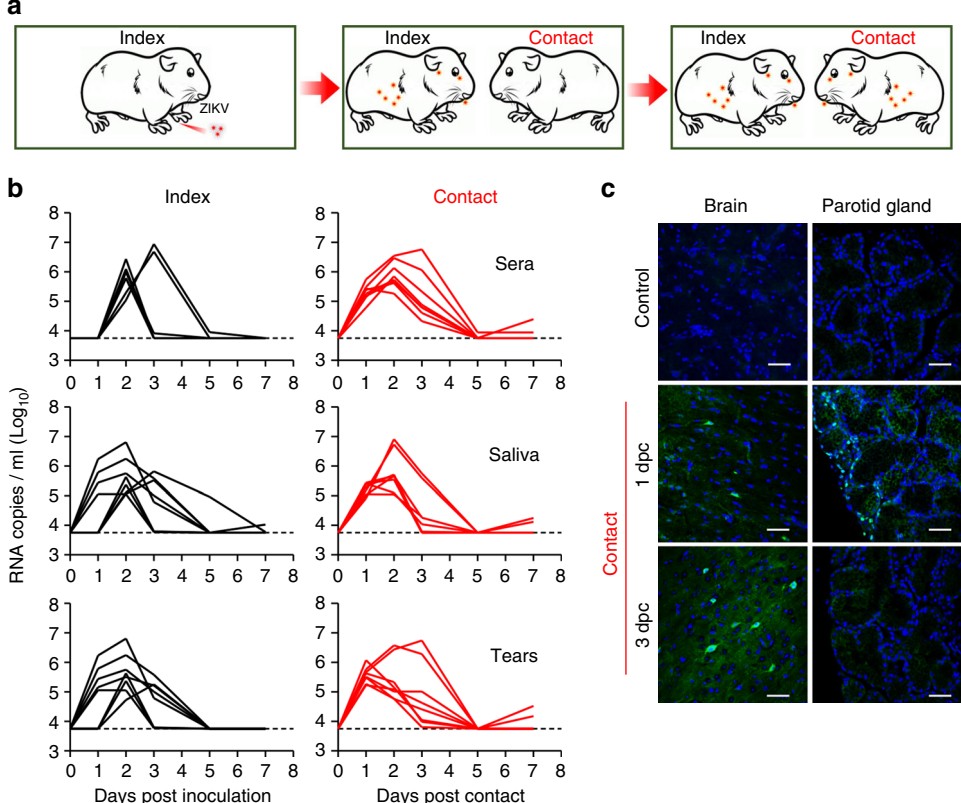

**Fig. 2** Close contact transmission of ZIKV in guinea pigs. **a** Schematic diagram of the close contact experiments. Briefly, each male guinea pig that received s.c. inoculation of ZIKV was placed in an isolated cage, then the other naïve male animal was co-caged at 1 or 3 dpi. **b** Serum, saliva, and tear samples were collected from the index ($n = 8$) and contact ($n = 8$) animals and subjected to viral load analysis. **c** Immunostaining of selected tissues from the contact animals at 1 and 3 dpc. Scale bar: 50 μm

and no obvious changes in body temperature or clinical symptoms were observed during the 14-day observation period. Significantly, sustained viremia was detected in all animals from 2 to 5 days post infection (dpi), with peak titers of 5.4 to 6.9 log RNA copies/ml at 3 dpi (Fig. 1a); a similar magnitude and pattern has also been observed in non-human primate models[10,11]. Specifically, we performed viral isolation on mosquito C6/36 cells using serum samples collected at 2 dpi, and the results showed that infectious ZIKV were directly recovered from three out of eight serum samples (Supplementary Table 1). We also detected robust viral RNA shedding in both saliva and tears, with a prolonged duration relative to that of viremia (Fig. 1a).

Then, necropsy of the inoculated guinea pigs was performed at 3 and 6 dpi, and high levels of ZIKV RNAs were detected in multiple organs at 3 dpi and decreased at 6 dpi (Supplementary Fig. 2). Immunostaining with ZIKV-specific antibodies revealed that substantial ZIKV-positive cells in the spleen, intestine, and testis at 3 dpi (Fig. 1b). Specifically, ZIKV antigens were predominantly found in the Leydig cells of the testis of guinea pigs, which is consistent with prior findings for mice[20,21]. Importantly, ZIKV infection resulted in substantial pathological changes in these organs. Characteristic changes included slight inflammatory cell infiltration and hemorrhage in the spleen and intestine, and destruction of the seminiferous epithelium with constricted tubules in the testis (Fig. 1c); these results were well correlated with immunostaining findings. Additionally, ZIKV-specific immunoglobulin G (IgG) antibodies were detected at 14 dpi and sustained thereafter (Supplementary Fig. 3). Overall, these results suggest that guinea pigs are susceptible to ZIKV infection and can be used to mimic potential contact transmission of ZIKV.

**ZIKV can be transmitted via close contact in guinea pigs**. We then tested the transmissibility of ZIKV by performing a direct contact experiment in the guinea pig model. Each male adult guinea pig (the index animal) was s.c. inoculated with $10^5$ PFU of ZIKV and raised in an individual cage. One or three days later, each infected animal was co-caged with a naïve male guinea pig (the contact animal) (Fig. 2a and Supplementary Fig. 4). As expected, viremia and viral secretion in saliva and tears were detected in all index animals ($n = 8$); remarkably, all contact guinea pigs (100%) readily developed viremia comparable in magnitude and duration to that of their index cage mates (Fig. 2b). Meanwhile, the viral shedding kinetics in saliva and tears of the contact animals were also comparable to that of the index animals (Fig. 2b). Especially, ZIKV viral RNAs were readily detectable in sera, saliva, and tears from all contact animals at 1 day post contact (dpc). Importantly, we detected robust ZIKV viral protein signals in the brain neurons of the contact guinea pigs at 1 and 3 dpc (Fig. 2c). Interestingly, ZIKV antigen-positive cells lining the striated ducts located close to the surface of the parotid glands were detected at 1 dpc, but disappeared at 3 dpc (Fig. 2c).

Then, full genome sequences of ZIKV were recovered from the ZIKV-positive parotid glands at 6 dpi by high-throughput sequencing. Sequence alignment to the original reference genome of ZIKV GZ01 revealed a panel of intra-host single-nucleotide variants (iSNV), with nucleotide substitution frequency above 5%[22], throughout the genome (Supplementary Fig. 5). In particular, an A to G substitution at nucleotide position 2287, resulting in an M469V substitution of the E protein, was identified (Supplemental Table 2). To further characterize the dynamics of genetic variations of M469V in ZIKV-infected guinea pigs, various tissues from the s.

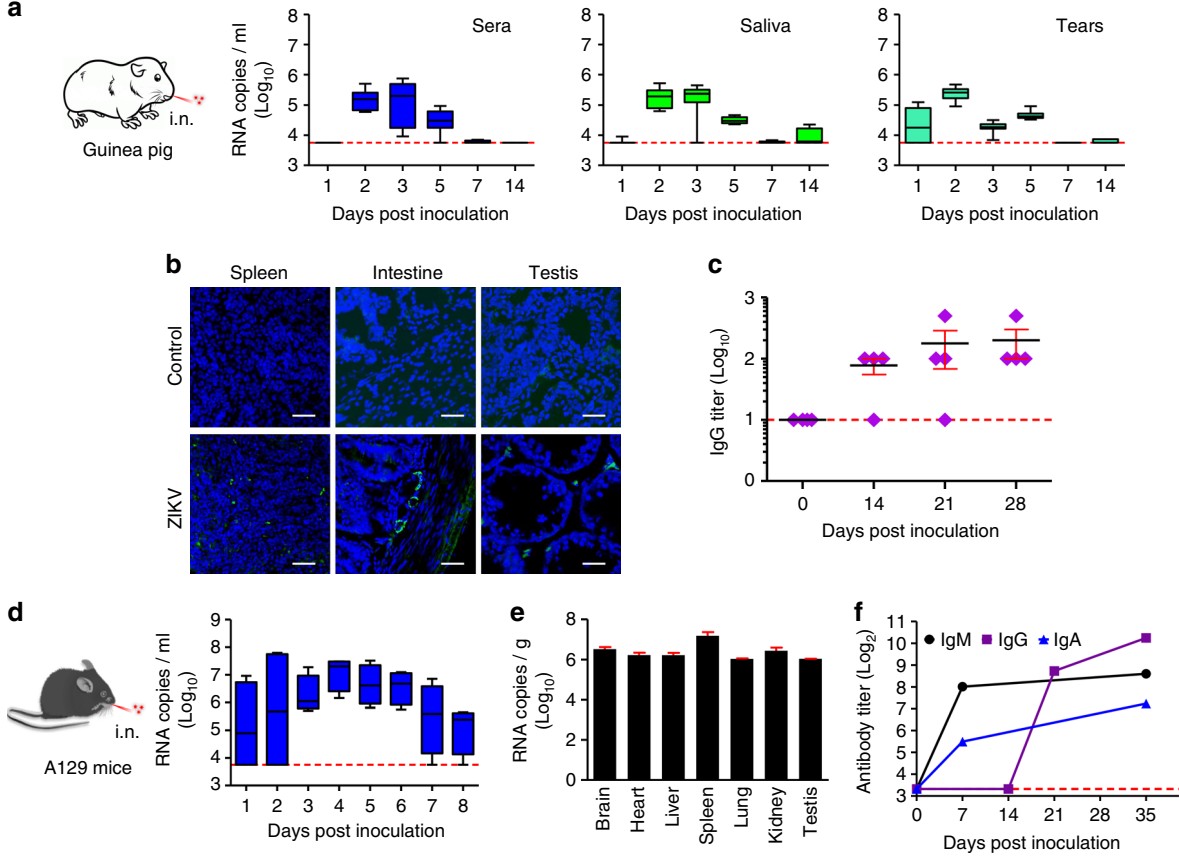

**Fig. 3** Guinea pigs and A129 mice are susceptible to ZIKV infection via an i.n. route. **a** Sera, saliva, and tears samples of four ZIKV-inoculated guinea pigs ($n = 4$) were collected following i.n. inoculation and analyzed as described above. **b** Immunostaining of indicated tissues were performed at 3 dpi. Scale bar: 50 µm. **c** ZIKV E protein-specific IgG antibody titers in guinea pigs ($n = 4$) were measured using ELISA. IgG antibody titers were calculated according to the highest reciprocal dilution of serum to give an OD greater than the sum of the background OD plus 0.01 units. The data is shown in mean ± SEM. The dotted lines represent the limits of detection of the ELISA assigned values of 10. **d** Viremia of four ZIKV-inoculated A129 mice ($n = 4$) at 1 to 7 days after i.n. inoculation was assessed. Dotted lines indicate the limit of detection. Whiskers: 5–95 percentile. **e** Tissue distribution of ZIKV RNAs in the i.n. inoculated A129 mice at 7 dpi. The data are shown as mean ± SD from two experimental replicates. **f** ZIKV E-specific IgA, IgM, and IgG antibody titers were measured using ELISA. The dotted lines represent the limits of detection of the ELISA assigned values of 10

c. inoculated animals were subjected to routine polymerase chain reaction (PCR)-based sequencing. The results showed that although the M469V substitution has not occurred in specific tissues (kidney and testis) at 3 dpi, all tested samples collected at 6 dpi have accumulated the M469V substitution (Supplementary Table 3).

Furthermore, we repeated the contact transmission experiment with a historical ZIKV strain, FSS13025, that was isolated in 2010 in Cambodia[23]. Again, we found 100% transmission efficiency in guinea pigs under the same experimental conditions, and all contact animals developed typical viral secretions in sera, saliva, and tears (Supplementary Fig. 6), although with lower magnitude and duration of viral shedding compared to those of ZIKV GZ01. Taken together, our results indicate that ZIKV was able to transmit from infected guinea pigs to the naïve co-caged animals by close contact.

**Intranasal infection of ZIKV in guinea pigs and A129 mice.** Further, we sought to clarify whether ZIKV infection can be established via an i.n. route in guinea pigs. Upon i.n. inoculation, typical viral shedding kinetics in sera, saliva, and tears were detected in all inoculated animals (Fig. 3a). ZIKV proteins were detected by immunostaining of selected organs, the spleen, intestine, and testis, at 3 dpi (Fig. 3b), and viral RNAs were also detected in multiple tested tissues at 3 and 6 dpi (Supplementary

Fig. 7). Additionally, ZIKV-specific seroconversion was seen at 14 dpi and maintained thereafter (Fig. 3c). Collectively, our results indicate that guinea pigs are susceptible to ZIKV infection via the i.n. route, and the robust viral secretions in saliva and tears, which might contribute to the observed contact transmission.

We also evaluated the infectivity of ZIKV via an i.n. route in the well-established A129 mouse model that are deficient in interferon-α/β receptor[24]. Groups of 4-week-old male A129 mice were subjected to i.n. inoculation with $10^5$ PFU of ZIKV, and all mice developed robust and persistent viremia (Fig. 3d). High levels of ZIKV RNAs were detected in multiple organs, including the brain, heart, liver, spleen, lung, kidney, and testis (Fig. 3e), which is consistent with previous findings from the s.c. inoculation[25]. Additionally, IgA and IgM antibodies against ZIKV were readily induced at 7 dpi, followed by IgG antibody responses (Fig. 3f). Together, these results indicate that potent nasal mucosal infection could be established via i.n. inoculation in A129 mice.

**Intranasal and intragastric infection of ZIKV in macaques.** Finally, to expand our findings to non-human primates, we inoculated adult cynomolgus macaques[11] with $10^5$ PFU of ZIKV via i.n. and i.g. routes (Fig. 4a). All macaques exposed to ZIKV via i.n. or i.g. inoculation ($n = 4$) developed obvious viremia within

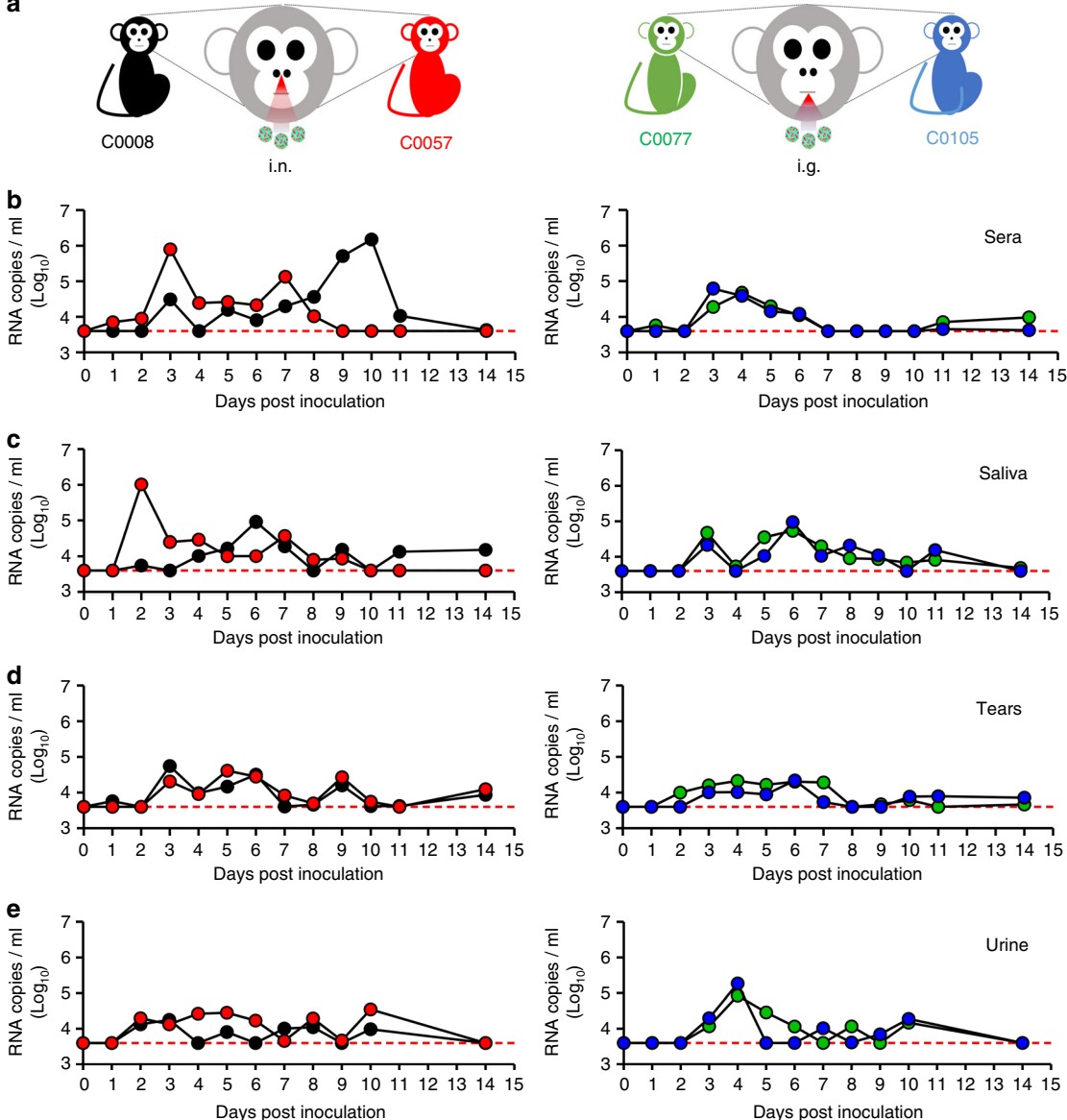

**Fig. 4** Characterization of ZIKV infection in cynomolgus monkeys following i.n. and i.g. inoculation. **a** Two adult cynomolgus monkeys received i.n. inoculation with $10^5$ PFU of ZIKV, and the other two animals received i.g. inoculation. Viral RNA loads in **b** sera, **c** saliva, **d** tears, and **e** urine were determined, respectively, at the indicated time points. The dotted lines indicate the limit of detection

15 dpi, and the duration and peak of viremia in the i.n. inoculated animals were greater than that in the i.g. inoculated animals (Fig. 4b). Meanwhile, robust ZIKV excretions were detected in saliva, tears, and urine in all infected animals (Fig. 4c–e), with a pattern similar to that observed in s.c. inoculated macaques[10,11,26]. In particular, viral shedding was delayed in the i.n. challenged animals compared with the i.g. challenged animals. Consistent with prior findings for s.c. inoculated animals[10,11,26], we observed no neurological symptoms or psychological abnormalities among the i.n. or i.g. inoculated macaques throughout the study. Blood chemistry analysis and complete blood cell counts revealed unified elevated aminotransferase (ALT) and creatinine (CREA) levels in all infected animals (Supplementary Fig. 8). Luminex assay of plasma from the ZIKV-infected cynomolgus macaques indicated that multiple proinflammatory cytokines, including interleukin-12 (IL-12), granulocyte–macrophage colony-stimulating factor, migration inhibitory factor, and IL-8, were elevated upon either i.n. or i.g.

infection (Supplementary Table 4). These results suggest that the i.n. or i.g. inoculation of ZIKV are capable of establishing systematic infection in cynomolgus monkeys.

## Discussion

Tremendous advances have been made to mimic ZIKV infection in animal models since 2016, which has significantly deepened our understanding of pathogenesis and boosted the development of antivirals and vaccines[27]. Our result showed that wild-type guinea pigs are susceptive to ZIKV infection upon various inoculation routes. Infectious ZIKV can be directly recovered from serum samples, ZIKV antigens were detected in multiple tissues, and ZIKV-specific antibodies were readily stimulated upon ZIKV inoculation in guinea pigs. Despite no clinical symptoms developed, viremia and viral secretions in saliva and tears well mimicked ZIKV replication in vivo. The magnitude and duration of viremia in guinea pigs was much greater than that in

immunocompetent mice[28], and the peak viremia was comparable to previous findings in human and non-human primate models[11,26,29]. During the submission of our manuscript, another paper[19] showed that Dunkin–Hartley strain guinea pigs developed viremia and seroconversion, as well as clinical symptoms, upon s.c. challenge with a higher dose ($10^6$ PFU) of ZIKV strain PRVABC59. Collectively, the guinea pig model provides an alternative choice for future pathogenesis study and antiviral drugs or preventive vaccine efficacy tests.

Our results clearly demonstrated that ZIKV infection could be efficiently established via i.n. or i.g. routes in A129 mice, guinea pigs, and non-human primates. Despite no clinical symptoms observed, all ZIKV-infected guinea pigs developed seroconversion and robust viral secretions in sera, saliva, and tears. Especially, ZIKV viral antigens were transiently detected in the parotid glands of infected guinea pigs. The parotid gland has been well recognized as a privileged site for immune evasion and persistence for many pathogenic viruses, including mumps virus, BK polyomavirus, human cytomegalovirus and Epstein–Barr virus, among others[30–32], most of which can be transmitted via contact with mucosal secretions, such as saliva, semen, and vaginal and cervical secretions. The presence of ZIKV antigens in the parotid glands at the early stage of ZIKV infection suggest that the oronasal mucosa may be a potential entry site that supports ZIKV replication. Interestingly, vaginal ZIKV infection of female mice leads to productive viral replication in the vaginal mucosa[4].

Especially, some preliminary in vitro assays have also demonstrated that human respiratory cells are highly susceptible to ZIKV[33]. Additionally, a recent paper published during the submission of our manuscript has demonstrated that direct inoculation of high-dose ZIKV into the tonsils in rhesus macaques resulted in productive infection[34]. These combined findings from cell culture and animal model call attention to the potential risk of ZIKV transmission via oral or nasal routes, such as the consumption of contaminated food or breast milk, or via nasal routes, such as the inhalation of or contact with contagious droplets or aerosols. In particular, individuals, especially pregnant women and health-care professionals, should be informed of the potential risk of direct contact with or oronasal exposure to ZIKV contaminants from symptomatic or asymptomatic patients.

Remarkably, our close contact transmission experiments in guinea pigs showed that the two ZIKV strains, FSS13025 and GZ01 isolated in 2010 and 2016, respectively, showed 100% transmission efficiency. All the naïve guinea pigs readily acquired ZIKV from their co-caged animals, and viral RNAs and antigens were present in multiple tissues of the contact animals. However, our experimental setup for contact transmission facilitates not only direct but also indirect contact, e.g., respiratory droplets or aerosols. Whether ZIKV was capable of transmitting via droplets or aerosol routes requires additional investigation with specific facility and animal models[35,36].

We also noted that the 2010 historical ZIKV strain FSS13025 was slightly attenuated in guinea pigs compared with the 2016 contemporary ZIKV strain GZ01 (Fig. 2b vs. Supplementary Fig. 6). The genetic mechanism behind these observations remains to be determined. Additionally, we identified a progressive adaption substitution (M469V) in the E protein during ZIKV infection in guinea pigs. Previously, bioinformatics analysis has also suggested the M469V amino acid substitution at E protein might contribute to the cross-species transmission between mammals and mosquitoes[37]. Further investigation with the well-established reverse genetic platforms of ZIKV[38,39] will help to characterize the potential genetic determinants of the infectivity and transmissibility of ZIKV in guinea pigs.

Finally, it remains unknown whether similar direct contact transmission modes observed in guinea pigs have occurred in human populations, and the contribution of contact transmission route during the explosive ZIKV pandemic remains elusive[40]. Very recently, infectious DENV have been recovered directly from respiratory specimens from patients with cough, rhinorrhea, and nasal congestion[41]. Direct transmission of DENV through mucocutaneous exposure to contagious blood from DENV-infected patients has also been reported[42]. Case reports of Zika infection that cannot be explained by mosquito biting or sexual behavior[43] are concerning. More evidence from both animal models and clinical settings are needed to distinguish close contact transmission from other known routes. Additionally, we have recently shown that contemporary ZIKV strains have acquired adaptive mutations that increased neurovirulence[44] and mosquito transmissibility[45]. Whether such mutations have some impact on its potential transmissibility via close contact warrant further investigation.

## Methods

**Ethics statement**. All animal experiments were performed under biosafety level-2 conditions and in strict accordance with the guidelines of the Chinese Regulations of Laboratory Animals (Ministry of Science and Technology of People's Republic of China) and Laboratory Animal-Requirements of Environment and Housing Facilities (GB 14925-2010, National Laboratory Animal Standardization Technical Committee). All procedures were approved by the Animal Experiment Committee of Laboratory Animal Center, AMMS, China (IACUC-13-2016-001).

**Cells and viruses**. BHK-21 cells (American Type Culture Collection (ATCC), #CCL-10) were cultured in Dulbecco's modified Eagle's medium (Invitrogen) with 10% fetal bovine serum (FBS), 100 U/ml of penicillin, and 100 μg/ml of strepto-mycin. C6/36 cells (ATCC #CRL-1660) were cultured in RPMI-1640 with 10% FBS, 100 U/ml of penicillin, and 100 μg/ml of streptomycin. The ZIKV GZ01 strain (GenBank accession no: KU820898) was isolated from a Chinese patient returned from Venezuela in 2016[6]. The FSS13025 strain (GenBank accession no: KU KU955593.1) was isolated in Cambodia in 2010[23]. Viral stocks were prepared in mosquito C6/36 cells, titrated in BHK-21 cell by plaque-forming assay, and stored as aliquots at −80 °C.

**Guinea pig experiments**. Groups of five-week-old male guinea pigs (Hartley strain) weighing ca. 300 g were obtained from Charles River Laboratories and raised in individual cages. Each animal was inoculated with the indicated dose of ZIKV strain GZ01 or FSS13025 by the s.c. or i.n. route. All animals were monitored daily for weight loss, body temperature change, and clinical symptoms. At 1, 2, 3, 5, 7, and 14 dpi, sera, salvia, and tears samples from ZIKV-inoculated animals were collected. Briefly, saliva samples were obtained by running a sterile swab under the animal's tongue. Tears samples were collected by gently running a sterile swab on the animal's eye. Swabs were placed immediately into 1.0 ml of viral transport medium (tissue culture medium 199 supplemented with 0.5% FBS and 1% anti-biotic/antimycotic) for 60 min. Samples were vortexed vigorously, then centrifuged for 10 min at 800×g before removing the swabs. Samples were stored at −80 °C until processing and for viremia. Viral loads of three samples were analyzed by quantitative reverse transcription PCR (RT-qPCR) and were expressed as RNA copies per milliliter. At 3 and 6 dpi (1 and 3 dpc), necropsy was performed and various tissues were harvested, weighed, and subjected to detection of viral load, or immunostaining, or histopathology assays, respectively.

The close contact transmission experiments were performed according to previous protocols[46,47]. Briefly, each male guinea pigs infected via s.c. inoculation was housed in individual cage, and the other naïve male guinea pig was co-caged at 1 or 3 dpi. Then, viral loads in sera, saliva, and tears samples were collected for viral load assays. Various tissues were collected at the indicated times for histopathology and immunostaining assays. Strict measures were followed to prevent aberrant cross-contamination.

**A129 mice experiments**. Groups of 4-week-old male A129 mice (Laboratory Animal Center, Academy of Military Medical Science) were inoculated with $10^5$ PFU of ZIKV GZ01 strain by the i.n. route. Clinical manifestation, morbidity, and mortality were monitored daily. Mice that were moribund or that lost >20% of starting weight were humanely killed. Serum samples were collected for viremia and antibody response at the indicated times. The brain, heart, liver, spleen, lung, kidney, intestine, and testis were collected at 7 dpi and subjected to viral load analysis.

**Monkey experiments**. Three-year-old female cynomolgus monkeys (weighing 3.5 to 4.5 kg) were pre-screened negative for IgG antibodies against ZIKV by enzyme-linked immunosorbent assay (ELISA). Then, two monkeys were i.n. or i.g. inoculated with $10^5$ PFU of ZIKV GZ01, respectively. After the inoculation, clinical

signs were recorded during 14-day observation period. Blood and major body fluids (saliva, tear and urine) were collected every day for determination of viral loads. Briefly, urine was collected from a container under the animal's cage, and was stored at –80 °C until analysis. Saliva was obtained by running a sterile swab under the animal's tongue. Tears were collected by gently running a sterile swab on the animal's eye. All samples were stored at –80 °C till use.

**Biochemistry and hematology analysis.** A panel of hematological parameters, i.e., white blood cell count, red blood cell count, hemoglobin, platelets, lymphocytes, monocytes, and neutrophils were analyzed in peripheral blood using a Celltac E MEK-7222 hematology analyzer (Nihon Kohden, Japan). Biochemical analysis, i.e., alanine aminotransferase (ALT), aspartate aminotransferase, total protein, albumin, glucose, urea, and CREA was assessed using a 7100 automated biochemical analyzer (Hitachi, Japan).

**Viral load assay.** Total RNA was extracted from 100 μl of serum and the grinded tissue supernatant using the PureLink® RNA Mini Kit (Life Technologies, USA) according to the manufacturer's recommendation and eluted in 60 μl of RNase-free water. Using ZIKV-specific primers and probe (ZIKV-ASF: 5′-GGTCAGCGTCCTCTCTAATAAACG-3′; ZIKV-ASR: 5′-GCACCCTAGTGTC-CACTTTTTCC-3′; ZIKV-Probe: 5′-FAM-AGCCATGACCGACACCACACCGT-BQ1-3′), RT-qPCR was carried out with the One-Step PrimeScript™ RT-PCR Kit (Takara, Japan) with the LightCycler system (Roche, USA). RNA copies per ml or RNA copies per gram were calculated from quantitative PCR Ct values as described previously[29].

**Virus isolation and identification.** Serum from infected guinea pigs at 1 dpi was inoculated in C6/36 mosquito cells and maintained in RPMI-1640 medium (Life Technologies, USA) supplemented with 2% FBA (Biowest, France) at 28 °C in 5% CO₂. After 6 dpi, viral RNAs were extracted from culture supernatant and detected by RT-qPCR as described above. RNA copies of culture supernatant increase at least 1000-fold compared to those of primary sample was deemed as positive for virus isolation.

**ZIKV-specific antibody detection.** Sera from A129 mice, guinea pigs, or monkeys were heat inactivated at 56 °C for 45 min before use. ELISA was performed to quantitate ZIKV E protein-specific IgA/IgG/IgM level as previously described[48]. Briefly, polysorb enzyme-linked immunosorbent assay plates (Nunc, USA) were coated with 50 ng per well of recombinat ZIKV E protein[49] diluted in phosphate-buffered saline (PBS) overnight at 4 °C. Plates were blocked in 5% (vol/vol) skimmed milk in PBS with 0.05% Tween-20 (PBST) for 1 h at 37 °C. Serum samples were serially diluted in PBST and added to wells for 1 h at 37 °C. The plates were then washed with PBST three times to remove unbound antibody. After washing, suitable concentration of horseradish peroxidase-conjugated anti-mice (ZSGB-Bio, China) or guinea pig IgA or IgM or IgG antibody (Abcam, UK) was added to the well for 1 h at 37 °C, and then the detection was performed using 3,3′,5,5′-tetramethylbenzidine substrate (Promega, USA). The ZIKV E-specific antibody titer was calculated according to the highest reciprocal dilution of serum to give an optical density (OD) greater than the sum of the background OD plus 0.01 units.

**Histopathology assay.** For histopathology, various tissue samples were collected from the mice or guinea pigs and fixed in 4% neutral-buffered formaldehyde, embedded in paraffin, sectioned, and stained with hematoxylin and eosin (H&E). Images were captured using Olympus BX51 microscope equipped with a DP72 camera.

**Immunofluorescence staining.** For immunostaining, various tissues were fixed in 4% paraformaldehyde for 24 h at 4 °C, and then dehydrated in 30% sucrose for 24 or 48 h. The fixed brains were frozen in tissue freezing medium and sectioned into 40 μm slices. The cyrosections were blocked at room temperature (RT) for 1 h in 3% bovine serum albumin, 10% FBS, and 0.2% Triton X-100 in PBS, and then incubated with ZIKV human convalescence serum (1:500 dilution) from a ZIKV patient described previously[50] at 4 °C overnight, and then washed with 0.2% Triton X-100 in PBS (3 × 10 min), followed by incubating in the fluorescein isothiocyanate-conjugated goat anti-human IgG (ZSGB-Bio, China) at RT for 1 h, and then washed for three times as described previously[51]. The convalescence serum from the ZIKV patient was obtained at Guangzhou Eighth People's Hospital with written informed consent, and the used of human sera was approved by the Institutional Review Board of Guangzhou Eighth People's Hospital. Nuclei were stained with 4′, 6-diamidino-2-phenylindole (DAPI, Invitrogen).

**Luminex assay.** Monkey sera collected at different time points were subjected for cytokine, chemokines, and growth factors analysis with the Monkey Cytokine Magnetic 29-Plex Panel (Thermo Fisher, LPC0005M, USA) according to the manufacturer's instruction. Each serum sample was measured with three replicates. The data were collected on Luminex200 and analyzed by Luminex xPONENT (Thermo Fisher, USA).

**ZIKV genome sequencing and analysis.** High-throughput sequencing of the parotid gland sample from the s.c. inoculated guinea pigs at 6 dpi was performed on an Illumina MiSeq sequencing machine. The genome sequences of ZIKV were assembled by mapping the reads to the reference genome of ZIKV GZ01. The mapping and iSNV detection were processed with CLC Genomic Workbench. The site with substitution frequency above 5% was considered as an iSNV[22].

For determination of consensus sequence containing the M469V substitution, RT-PCR were performed with RNAs extracted from various tissue samples from the ZIKV-infected guinea pigs. The primers used for RT-PCR and Sanger sequencing include: ZIKV-F3 (1420–1440): 5′-GGAAGCCTAGGACTTGATTGT-3′; ZIKV-R3(2376-2397): 5′-CGAGCACCCCACATCAGCAGAG-3′. Sequence fragments were assembled into a consensus sequence with DNA STAR software, version 7.0.

**Data availability.** All sequence data generated by high-throughput sequencing was deposited in Sequence Read Archive with accession codes SAMN07780900, SAMN07780901, SAMN07780902, and SAMN07780903, respectively. All relevant data are available from the authors on request.

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

## Acknowledgements

We thank Prof. Jing An (Capital Medical University, China) for helpful discussion, and the veterinarians from the Laboratory Animal Center at the Academy of Military Medical Sciences for their excellent technical support. This work was supported by the National Natural Science Foundation of China (NSFC) (Nos. 81772176, 31770190, and 81661148054), the National Key Research and Development Project of China (Nos. 2016YFD0500304, 2016YFC1201000, and 2016ZX10004001-008) the National Science and Technology Major Project of China (Nos. 2017ZX09101005 and 2017ZX10304402), the Special Program of Guangdong Provincial Department of Science and Technology (No. 2016A020248), and the Guangzhou Science and Technology Program for Public Wellbeing (Nos. 201508020263 and 201704020229). C.-F.Q. was supported by the Excellent Young Scientist Program (No. 81522025), the Innovative Research Group of the NSFC (No. 81621005), and a Newton Advanced Fellowship from the UK Academy of Medical Sciences (No. NAF003/1003). Z.X. was supported by the NSFC (Nos. 31730108, and 31430037).

## Author contributions

C.-F.Q. and Y.-Q.D. conceived and designed this study. Y.-Q.D., N.-N.Z., X.-F.L., M.T., Y.-Q.W., Y.-F.Q., H.-L.D., J.-W.F., X.-Y.H., H.-L.D. and H.F. performed the experiments and analyzed the data. J.-N.H., H.F., Y.-G.W., Y.-G.T., Z.X. and F.-C.Z. contributed reagents and data analysis. C.-F.Q., Y.-Q.D., X.-F.L. and Z.X. wrote the paper with contributions from all authors.

## Additional information

**Competing interests:** The authors declare no competing financial interests.

