## [Peer Review File · Nature Communications]

Reviewers' comments:

Reviewer #1 (Remarks to the Author):

Title: Intranasal infection with and contact transmission of Zika virus in guinea pigs and non-human primates.

Review: Zika virus (ZIKV) is a mosquito transmitted Flavivirus that is shed in a number of bodily fluids including saliva, urine, and tears. Close contact transmission has been reported but the specifics of this mode of transmission have not yet been rigorously tested. In the current manuscript the authors determined the ability of guinea pigs to be used as a ZIKV transmission animal model. Guinea pigs (gp) are readily infected subcutaneously, become viremic and shed virus in saliva and tears. Virus was detected in tissues from the gp. Importantly, naïve gp became infected when co-housed with infected gp. Intranasal inoculation also resulted in infection of gp as well as mice and cynomolgus macaques. The authors suggest that this work highlights the risk of ZIKV transmission via close contact.

Concerns: Overall the results presented in the manuscript are interesting and on point with characterizing an important feature of ZIKV transmission. However, there are a number of concerns that dampen overall enthusiasm for this manuscript.

1. Figure 1, the authors show virus staining and vRNA detection in spleen, intestine and testis. What other organs were tested? The authors should add the data for these additional organs even if negatives so that we can determine what extent of viral dissemination. Similarly, was virus detected in the urine from the infected animals?
2. The authors sequenced virus recovered in the parotid glands; however, specific details and interpretation of the results are lacking. Were the mutations present in the inoculum and if so at what frequency? What is the frequency of each mutation present in the in vivo samples? Importantly, is there a functional biologic outcome for these mutations? One would expect that if the mutations adapted the virus to gp that the virus would have a growth advantage; was this formally tested?
3. The experimental information for the co-housing gp experiment are lacking, which makes it difficult to interpret their findings. For example, what was the sex of the gp used in this experiment? The last sentence of the first paragraph on page 5 reads "likely through direct contact with contagious viral secretions in saliva." However, there is no proof that the virus was transmitted via saliva vs. urine, tears or through sexual contact. In order to suggest that it is through saliva, the authors should test whether there is infectious virus in the saliva and whether this can be used to directly infect naïve animals. The data shown reflect vRNA loads and not infectious virus.
4. The authors demonstrate that subcutaneous infection of gp with Strain GZ01 results in protracted viremia, is this finding statistically significant?
5. Paragraph 1 on Page 5: Figure S4 and S5 are mislabeled.

Reviewer #2 (Remarks to the Author):

Yong-Qiang Deng et al. report a characterization of the guinea pig as a model host for Zika virus. Following up on the observation that ZIKV transmits between co-caged guinea pigs efficiently, they additionally performed a series of experiments in guinea pigs, A129 mice and cynomolgus macaques aimed at determining whether ZIKV is infectious following oral and nasal routes of entry. The results indicate that guinea pigs are susceptible to ZIKV infection by sub-cutaneous and intra-nasal routes, with viral RNA detected in tears, saliva and serum and viral antigen detected in multiple tissues. With both a recent ZIKV isolate and a historical strain, highly efficient transmission between co-caged animals was noted. Although ZIKV is typically transmitted by mosquito bite to humans, this observation in guinea pigs suggests that transmission via ingestion, inhalation or contact of nasal / oral mucosa with contaminated fomites may occur. To pursue this concept further, the authors tested direct inoculation by nasal or oral routes in guinea pig, mouse and macaque models. Infection was productive via IN inoculation in mice and guinea pigs and by IN and IG inoculation in macaques. The authors conclude that ZIKV may transmit in humans via non-vector borne routes (other than sexual transmission), including by close contact, ingestion of contaminated materials, and inhalation of droplets or aerosols.

This is a solid study, which clearly demonstrates highly efficient transmission of ZIKV between co-caged guinea pigs. While the significance to humans is challenging to verify for obvious reasons, the authors have followed a logical approach of determining whether IN and/or IG routes of inoculation allow productive ZIKV infection in two additional animal models. Results suggest that the observation in guinea pigs is generalizable to multiple species. In my opinion, one major question has been left unanswered: in quantitative terms, what is the relative efficiency of infection via sub-cutaneous and intranasal routes?

Specific comments:

1. As noted above, the conclusions of the paper would be strengthened, and important insight into relative importance of different transmission routes would be obtained, by determining the median infectious dose of guinea pigs by SC, IN (and IG) routes.
2. All viral growth data are reported as RNA copies per ml. Data is needed on the relationship between an infectious unit and RNA copies in each type of sample analyzed. In addition, information on how RNA copies per ml were calculated from qPCR Ct values is needed (e.g. was a standard curve generated?).
3. A sample of the virus stock used for inoculation should be sequenced to determine whether SNVs were present initially or arose within the animal.
4. Page 4, lines 22-23. Demonstration of a low GPID50 is needed to support the statement that GPs are "highly susceptible to ZIKV infection".
5. Page 4 line 2 and page 5 line 15: please give dose used.
6. X-Axes of Fig.2 and Supplementary Figs. 4 and 5 are unclear with respect to the contact animals. Is day 0 the day on which inoculated animals were inoculated, or the day on which contact was initiated?
7. The fact that A129 mice are deficient in IFNAR should be stated in the main text.
8. please define iSNV

Reviewers' comments:

Reviewer #1 (Remarks to the Author):

Title: Intranasal infection with and contact transmission of Zika virus in guinea pigs and non-human primates.

Review: Zika virus (ZIKV) is a mosquito transmitted Flavivirus that is shed in a number of bodily fluids including saliva, urine, and tears. Close contact transmission has been reported but the specifics of this mode of transmission have not yet been rigorously tested. In the current manuscript the authors determined the ability of guinea pigs to be used as a ZIKV transmission animal model. Guinea pigs (gp) are readily infected subcutaneously, become viremic and shed virus in saliva and tears. Virus was detected in tissues from the gp. Importantly, naïve gp became infected when co-housed with infected gp. Intranasal inoculation also resulted in infection of gp as well as mice and cynomolgus macaques. The authors suggest that this work highlights the risk of ZIKV transmission via close contact.

Concerns: Overall the results presented in the manuscript are interesting and on point with characterizing an important feature of ZIKV transmission. However, there are a number of concerns that dampen overall enthusiasm for this manuscript.

Response: Thanks for the summary and positive comments.

1. Figure 1, the authors show virus staining and vRNA detection in spleen, intestine and testis. What other organs were tested? The authors should add the data for these additional organs even if negatives so that we can determine what extent of viral dissemination. Similarly, was virus detected in the urine from the infected animals?

Response: Thanks for the comments. We have repeated the s.c. infection experiments, and multiple organs, including the brain, heart, liver, spleen, lung, kidney, intestine, and testis were harvested at day 3 and 6 from the infected animals. RT-qPCR results showed that vRNAs were detected in all tested organs (shown in the new supplemental Fig. 2), indicating systematic infection have been established, which was consistent with previous findings from A129 mice and non-human primates. In our study, we selected the spleen, intestine and testis, that are well investigated in other animal models, for further immunostaining and pathology assays.

We agree with the reviewer that the viral excretion in urine is interesting and has been well investigated in other animal models and humans. While unlike murine or monkey, urine samples are quite difficult to collect from guinea pigs due to technical reasons. Not like monkeys,

consistent urine samples collected by a pallet under the cage is impossible due to limited volume. We have also tried to establish the urethral catheterization using a gauge venous catheter as previously described (Murdoch, et al. Eur J Clin Microbiol Infect Dis., 1999), while our veterinary officers thought this protocol will cause severe damage to the animals. Although we did collect a few positive urine samples, yet that is far from reaching any conclusion. Thus, the viral secretions in ZIKV-infected guinea pigs remains unknown at present.

2. The authors sequenced virus recovered in the parotid glands; however, specific details and interpretation of the results are lacking. Were the mutations present in the inoculum and if so at what frequency? What is the frequency of each mutation present in the in vivo samples? Importantly, is there a functional biologic outcome for these mutations? One would expect that if the mutations adapted the virus to gp that the virus would have a growth advantage; was this formally tested?

Response: *Thanks for the suggestion. We feel sorry for not providing enough details because the MS was prepared as Report for Nature Medicine and transferred to Nature communications. All mutations were identified by high-throughput sequencing and compared with the original genome sequence of ZIKV stocks. The global distribution and frequency of each nucleotide substitution was shown in the **new supplementary Fig. 5 and supplemental Table 2**. Specifically, we characterized the dynamics of genetic variations of the amino acid mutation M469V in guinea pigs by routine PCR-based sequencing, revealed the M469V mutation occurred in most organs except the kidney and testis at 3 dpi, while all tested samples collected at 6 dpi have accumulated the M469V mutation (shown in **new supplementary Table 3**). All these details have now been combined in the results and materials and methods sections.*

We agree with the reviewer that viral evolution within the host is an important question, particularly as animal models are developed to study pathogenesis, as well as antiviral and vaccine development. For the biological outcome for these mutations, further studies using reverse genetic technology of ZIKV are needed to elucidate the sequential acquisition of these mutations and their individual contributions to pathogenesis. Actually, the M469V mutation observed in our experimental model has been mentioned by previous bioinformatic analysis (Wang et al. Cell Host & Microbe, 2016), the corresponding role during host adaptation deserves further investigation. We believe this will be another good story, and we have discussed this issue in the discussion.

3. The experimental information for the co-housing gp experiment are lacking, which makes it

difficult to interpret their findings. For example, what was the sex of the gp used in this experiment? The last sentence of the first paragraph on page 5 reads “likely through direct contact with contagious viral secretions in saliva.” However, there is no proof that the virus was transmitted via saliva vs. urine, tears or through sexual contact. In order to suggest that it is through saliva, the authors should test whether there is infectious virus in the saliva and whether this can be used to directly infect naïve animals. The data shown reflect vRNA loads and not infectious virus.

Response: *Thanks for the comments. We feel sorry for not providing enough information in last version. The close contact transmission experiments in guinea pigs were performed according to previous protocols for influenza virus (Bouvier, J Virol, 2008; Pica, et al. J Virol, 2012). We have now provided more details about the close contact transmission experiments in the materials and methods section. All animals used are male to exclude the possibility of sexual transmission. Further, we agree with the reviewer that our original remark "that saliva as contagious source" was not accurate enough, and now we delete the sentence accordingly. In fact, the accurate transmission source could be complex, maybe saliva, tears, nasal secretions, or blood due to biting, fighting, or other actively interaction between the two animals. Also, like previous findings for other pathogenic virus, the contact experiments involve direct contact and indirect contact, whether airborne or droplet transmission have occurred depends on further extensive experiments. Whatever, we have changed the corresponding remarks about contiguous source and extensively discussed this issue in our revision. Furthermore, thanks for the suggestion for direct inoculation of saliva, while this experiment is technically not practical. Because the total volume of saliva that containing complex mixture (not pure media) is limited, that is far from establishing real infection in animals. Even for influenza A virus, there is no evidence showing that direct inoculation of clinical samples could lead to infection.*

4. The authors demonstrate that subcutaneous infection of gp with Strain GZ01 results in protracted viremia, is this finding statistically significant?

Response: *Based on our data, the viremia peak titer in the SC-inoculated animals is obviously higher 10-fold than that of IN-inoculated animals (Fig.2a and 3b). The p values obtained using the parametric two-tailed unpaired t test with Welch's correction have significant difference (***) ($p < 0.001$). We did not note obvious difference in viremia duration in animals inoculated via different routes.*

5. Paragraph 1 on Page 5: Figure S4 and S5 are mislabeled.

Response: *Corrected.*

Reviewer #2 (Remarks to the Author):

Yong-Qiang Deng et al. report a characterization of the guinea pig as a model host for Zika virus. Following up on the observation that ZIKV transmits between co-caged guinea pigs efficiently, they additionally performed a series of experiments in guinea pigs, A129 mice and cynomolgus macaques aimed at determining whether ZIKV is infectious following oral and nasal routes of entry. The results indicate that guinea pigs are susceptible to ZIKV infection by sub-cutaneous and intra-nasal routes, with viral RNA detected in tears, saliva and serum and viral antigen detected in multiple tissues. With both a recent ZIKV isolate and a historical strain, highly efficient transmission between co-caged animals was noted. Although ZIKV is typically transmitted by mosquito bite to humans, this observation in guinea pigs suggests that transmission via ingestion, inhalation or contact of nasal / oral mucosa with contaminated fomites may occur. To pursue this concept further, the authors tested direct inoculation by nasal or oral routes in guinea pig, mouse and macaque models. Infection was productive via IN inoculation in mice and guinea pigs and by IN and IG inoculation in macaques. The authors conclude that ZIKV may transmit in humans via non-vector borne routes (other than sexual transmission), including by close contact, ingestion of contaminated materials, and inhalation of droplets or aerosols.

This is a solid study, which clearly demonstrates highly efficient transmission of ZIKV between co-caged guinea pigs. While the significance to humans is challenging to verify for obvious reasons, the authors have followed a logical approach of determining whether IN and/or IG routes of inoculation allow productive ZIKV infection in two additional animal models. Results suggest that the observation in guinea pigs is generalizable to multiple species. In my opinion, one major question has been left unanswered: in quantitative terms, what is the relative efficiency of infection via sub-cutaneous and intranasal routes?

Response: *Thanks for the very positive comments and nice suggestion. In our experiments, we used the same dose of ZIKV infection, thus we can directly compare the relative efficiency of infection by SC and IN routes in various animal models. As shown in original Fig.2a and 3b (see below), we can clearly see that the viral loads in sera and saliva from the SC-infected guinea*

pigs were obviously higher compared to those of the IN-inoculated animals, which indirectly suggesting infection by SC route is relatively more effective than that by the IN route. Consistently, results from A129 mice and monkeys also indicated that the SC route infection seemed led to more robust viremia and viral excretion in various body fluids. We have mentioned this point in the discussion in our revision.

Figure 1. Comparison of relative infection efficiency of ZIKV by the SC and IN routes

Specific comments:

1. As noted above, the conclusions of the paper would be strengthened, and important insight into relative importance of different transmission routes would be obtained, by determining the median infectious dose of guinea pigs by SC, IN (and IG) routes.

Response: Thanks for the constructive suggestion. We have now performed additional experiments to determine the median infectious dose of ZIKV in guinea pigs. Briefly, 10-fold dilutions of ZIKV GZ01 strain between 10^5 to 10^2 PFU were inoculated to male guinea pigs (Hartley strain) weighing 300 g (four per dilution) by the SC route. The animals were monitored for survival, weight loss, and clinical signs of disease. Blood of ZIKV-infected guinea pig was collected and analyzed by RT-qPCR for measuring viremia. As shown below, the percentage of viremia positive animals was dose-dependent manner, 100% (4/4), 75% (3/4), 25% (1/4) of animals inoculated with 10^5 , 10^4 and 10^3 developed obvious viremia, while no animals inoculated with 10^2 PFU developed viremia. The mean infectious dose was calculated to 3.5 (Log10) PFU by the Reed–Muench method. We have added these results as **a new supplemental Fig.1** in our revised manuscript. Previously, 10^3 PFU of ZIKV can lead to 100% infection rate in A129 mice and several species of macaques (Dowall SD, et al. PLoS Negl Trop Dis, 2016; Li XF, et al. EBioMedicine, 2016; Dudley DM, et al. Nat Commun, 2016). Thus, both SC and IN routes

represents effective infection routes, and the IN route is not so effective as the SC route. We have mentioned this issue in our revision.

Figure 2. The dose-dependent infection of ZIKV in guinea pigs

2. All viral growth data are reported as RNA copies per ml. Data is needed on the relationship between an infectious unit and RNA copies in each type of sample analyzed. In addition, information on how RNA copies per ml were calculated from qPCR Ct values is needed (e.g. was a standard curve generated?).

Response: *Thanks for the comments. Despite viral RNA copies can not translate into the infectious titer unit PFU directly, experience has indicated flavivirus viral RNA copies overestimated PFUs by ~2-3 logs. In fact, previous study has also founded that the RNA copy number/PFU ratio for ZIKV is $(1.03 \pm 0.08) \times 10^3$ in pure media (Xie XP, et al. MBio, 2017). Due to various tissue samples are assayed in our study, we didn't analyze the relationship between an infectious unit and RNA copies in each type of sample. Actually, our in-house RT-qPCR assay have been well used in our many previous publications (Li XF, et al. EBioMedicine, 2016; Li C, et al. Immunity, 2016; Li C, et al. Cell Stem Cell, 2016; Li Z, et al. Cell Res, 2017; Liu ZY, et al. J Virol. 2017).*

*During revision, we have tried to isolate infectious virus in C6/36 cells directly from the serum samples with high viral RNA loads, and after two passages, the RNA copies increased at least 1000-fold in 3 of 8 serum samples. The results also well demonstrated the presence of infectious ZIKV in serum. This result has been combined as a **new supplemental Table 1**.*

For information on how RNA copies per ml were calculated from qPCR Ct values as described previously (Li XF, et al. EBioMedicine, 2016). Briefly, the Xho I-linearized plasmid containing

full-length genomic cDNA clone of ZIKV strain GZ01/2016 was subjected to in vitro transcription using RiboMAX Large Scale RNA Production System. The RNA transcript was purified using PureLink RNAmimi kit (Life Technology) according to the manufacturer's instructions and quantified using spectrophotometry on Nanodrop®. The purified RNA was diluted 10-fold serially using RNases-free water and was detected using quantitative real-time reverse transcriptase PCR (qRT-PCR). Threshold cycle (Ct) values for the known concentrations of the RNA were plotted against the log of the number of genome equivalent copies. The resultant standard curve was used to determine the number of genome equivalents of ZIKV RNA in samples. The determination of the detection limit was based on the lowest level at which viral RNA was detected and remained within the range of linearity of a standard curve (Ct values < 38). We have cited the corresponding reference and amended these details in the material and methods accordingly.

3. A sample of the virus stock used for inoculation should be sequenced to determine whether SNVs were present initially or arose within the animal.

Response: *Thanks for the reviewer's comment. Yes, all iSNVs was identified by comparing with the original sequence of the virus stock used for inoculation. Additionally, we also compared the virial sequences from different samples collected at 3 and 6 dpi, and the results showed that the M469V mutation did not occur in some specific tissues at 3 dpi, but completely substituted at 6 dpi, supporting a dynamic adapting process.*

4. Page 4, lines 22-23. Demonstration of a low GPID50 is needed to support the statement that GPs are "highly susceptible to ZIKV infection".

Response: *Thanks for the comments. As responded above, based on our additional experiment for mean infectious dose and prior publications, we have changed "highly susceptible" into "susceptible" accordingly.*

5. Page 4 line 2 and page 5 line 15: please give dose used.

Response: *Dose included.*

6. X-Axes of Fig.2 and Supplementary Figs. 4 and 5 are unclear with respect to the contact animals. Is day 0 the day on which inoculated animals were inoculated, or the day on which contact was initiated?

Response: *Thanks for the reviewer's advice. All improved.*

7. The fact that A129 mice are deficient in IFNAR should be stated in the main text.

Response: *Detail included.*

8. please define iSNV

Response: *The have included the refencece (Ni et al. Nat Microbiol. 2016), and include basic information in the material and methods section.*

REVIEWERS' COMMENTS:

Reviewer #1 (Remarks to the Author):

The authors have adequately addressed my previous concerns. Additional experimental data is provided to determine viral tissue distribution in the Guinea pig model. Sequence information is now provided in more detail.

However, there is still one point that should be made. While it might be difficult to collect urine at specific time points, urine could be directly collected from the bladder at the time of necropsy. This should be considered for future studies.

The study represents the development of a guinea pig model of ZIKV infection and offers an interesting look at transmission.

Reviewer #2 (Remarks to the Author):

The authors were largely responsive to the reviewers' comments and the revisions have strengthened the manuscript. The description of a guinea pig model for ZIKV infection and transmission, and the novel insights into potential modes of transmission reported, will be of interest to a broad audience.

Reviewer #1 (Remarks to the Author):

The authors have adequately addressed my previous concerns. Additional experimental data is provided to determine viral tissue distribution in the Guinea pig model. Sequence information is now provided in more detail.

However, there is still one point that should be made. While it might be difficult to collect urine at specific time points, urine could be directly collected from the bladder at the time of necropsy. This should be considered for future studies.

The study represents the development of a guinea pig model of ZIKV infection and offers an interesting look at transmission.

Reply: Thanks for the comments!

Reviewer #2 (Remarks to the Author):

The authors were largely responsive to the reviewers' comments and the revisions have strengthened the manuscript. The description of a guinea pig model for ZIKV infection and transmission, and the novel insights into potential modes of transmission reported, will be of interest to a broad audience.

Reply: Thanks for the comments!